# Pathogenic *Escherichia coli* Possess Elevated Growth Rates under Exposure to Sub-Inhibitory Concentrations of Azithromycin

**DOI:** 10.3390/antibiotics9110735

**Published:** 2020-10-26

**Authors:** Tran Tuan-Anh, Ha Thanh Tuyen, Nguyen Ngoc Minh Chau, Nguyen Duc Toan, Tran Hanh Triet, Le Minh Triet, Nguyen Hoang Thu Trang, Nguyen Thi Nguyen To, Josefin Bartholdson Scott, Hao Chung The, Duy Pham Thanh, Hannah Clapham, Stephen Baker

**Affiliations:** 1The Hospital for Tropical Diseases, Wellcome Trust Major Overseas Programme, Oxford University Clinical Research Unit, Ho Chi Minh City Q5, Vietnam; anhtt@oucru.org (T.T.-A.); tuyenht@oucru.org (H.T.T.); chaunnm@oucru.org (N.N.M.C.); toannd@oucru.org (N.D.T.); trietlm@oucru.org (L.M.T.); trangnht@oucru.org (N.H.T.T.); tontn@oucru.org (N.T.N.T.); haoct@oucru.org (H.C.T.); duypt@oucru.org (D.P.T.); 2Division of Aquacultural Biotechnology, Biotechnology Center of Ho Chi Minh City, Ho Chi Minh City Q5, Vietnam; trietht@dab.vn.org; 3Department of Medicine, Cambridge Institute of Therapeutic Immunology & Infectious Disease (CITIID), University of Cambridge, Cambridge CB2 0AW, UK; jb2143@medschl.cam.ac.uk; 4Saw Swee Hock School of Public Health, National University of Singapore, Singapore 117549, Singapore; hannah.clapham@nus.edu.sg

**Keywords:** *Escherichia coli*, azithromycin, tetracycline, growth rate, fitness cost, antimicrobial resistance, resistome, morphology, physiology

## Abstract

Antimicrobial resistance (AMR) has been identified by the World Health Organization (WHO) as one of the ten major threats to global health. Advances in technology, including whole-genome sequencing, have provided new insights into the origin and mechanisms of AMR. However, our understanding of the short-term impact of antimicrobial pressure and resistance on the physiology of bacterial populations is limited. We aimed to investigate morphological and physiological responses of clinical isolates of *E. coli* under short-term exposure to key antimicrobials. We performed whole-genome sequencing on twenty-seven *E. coli* isolates isolated from children with sepsis to evaluate their AMR gene content. We assessed their antimicrobial susceptibility profile and measured their growth dynamics and morphological characteristics under exposure to varying concentrations of ciprofloxacin, ceftriaxone, tetracycline, gentamicin, and azithromycin. AMR was common, with all organisms resistant to at least one antimicrobial; a total of 81.5% were multi-drug-resistant (MDR). We observed an association between resistance profile and morphological characteristics of the *E. coli* over a three-hour exposure to antimicrobials. Growth dynamics experiments demonstrated that resistance to tetracycline promoted the growth of *E. coli* under antimicrobial-free conditions, while resistance to the other antimicrobials incurred a fitness cost. Notably, antimicrobial exposure heterogeneously suppressed bacterial growth, but sub-MIC concentrations of azithromycin increased the maximum growth rate of the clinical isolates. Our results outline complex interactions between organism and antimicrobials and raise clinical concerns regarding exposure of sub-MIC concentrations of specific antimicrobials.

## 1. Introduction

Antimicrobial resistance (AMR) in bacterial pathogens is a well-recognized global health problem, causing significant mortality and morbidity. In 2016, 58.6% of *Escherichia coli (E. coli)*, 34.5% of *Klebsiella pneumoniae (K. pneumoniae)*, 33.9% *Pseudomonas aeruginosa*, 55.4% of *Acinetobacter spp.*, and 19.8% of *Staphylococcus aureus* isolates in Europe were resistant to at least one antimicrobial [1]. The situation in low- and middle-income countries (LMICs), such as Vietnam, may be worse, where approximately 50% of *E. coli* [2] and 91% of *Shigella flexneri* [3] isolates from clinical samples were resistant to at least three antimicrobial classes (multi-drug-resistant (MDR)). According to WHO, hospitalization costs for patients infected with drug-resistant bacteria are significantly higher than that for patients infected with susceptible bacteria [4]. Indeed, it is estimated that resistant pathogens may result in a total economic cost of USD 0.5 billion and USD 2.8 billion in Thailand and the US, respectively [5]. The outlook is bleak, as the discovery and development of new antimicrobials have declined dramatically [6]. Hence, there is an urgent need for novel approaches to comprehend how AMR organisms emerge and spread and understand how antimicrobials impact on bacterial cells.

*E. coli* is a member of the Enterobacteriaceae and a component of the healthy human gut microbiota. *E. coli* has been reported as the most prevalent pathogen associated with bacteraemia in Asia (43.8%) [2] and urinary tract infections in Africa (41.9%) [7]. In South America, diarrheagenic *E. coli* was responsible for 45.2% of diarrheal cases in children under five years [8]. Particularly, ETEC (enterotoxigenic *E. coli*) is estimated to cause approximately 222 million diarrheal cases and 51,200 deaths globally in 2016 [9]. The high prevalence and burden of *E. coli* has made it one of the most important pathogens and one of the best studied group of microorganisms in science. However, we have limited contemporary data on the impact of antimicrobials on MDR *E. coli* associated with disease. A scenario may be all too familiar in LMICs where empirical antimicrobials are administered to patients without any microbiological diagnostics or antimicrobial susceptibility testing.

Although antimicrobial susceptibility testing is the gold standard approach to identify and describe resistant bacteria, it provides no insight into the molecular and cellular response triggered by antimicrobial exposure. There have been previous investigations into the molecular mechanisms of AMR in specific bacteria [2] and their morphological [10,11,12] and physiological characteristics [13] under antimicrobial treatments. However, there are few studies directly linking molecular mechanisms of AMR with the morphology and physiology of antimicrobial-resistant organisms. Given the high burden of resistant *E. coli* in LMICs, we designed a holistic investigation to provide an in-depth understanding of AMR in bloodborne *E. coli* from both a cellular and molecular perspective using microscopic techniques and whole-genome sequencing. Additionally, we studied the growth dynamics of pathogenic *E. coli* to measure the physiological changes of *E. coli* with a diverse compendium of AMR genes upon short-term exposure to different antimicrobial classes.

## 2. Results

### 2.1. Resistome and Morphology Associated with AMR

The 27 clinical *E. coli* isolates exhibited a high prevalence of AMR phenotypes, with all isolates resistant to at least one antimicrobial (Table 1). Among those, 70.4%, 70.4%, 88.9%, 74.1%, and 64.0% were resistant to ciprofloxacin, azithromycin, ceftriaxone, tetracycline, and gentamicin, respectively (Table 1). Additionally, 22/27 (81.5%) isolates were resistant to at least three different classes of antimicrobials and defined as MDR [14].

To investigate the AMR gene profiles of the 28-isolate collection, WGS was performed on the 27 clinical *E. coli* isolates and a whole-genome sequence of *E. coli* ATCC 25922 was obtained from the NCBI database. The most prevalent sequence type (ST) was ST1193, which accounted for 8/27 isolates (29.6%); the remainder fell into 11 different STs (Figure 1b). A resistome analysis demonstrated that resistance to azithromycin and tetracycline was associated with the presence of *mphA* and the *tet* gene variants (*tetA/tetR* or *tetB*), respectively. All isolates harboring *aac(3)-IIa* were resistant to gentamicin. In contrast, the resistome data were insufficient for explaining ciprofloxacin resistance, which is commonly attributed to mutations in the Quinolone resistance determining region (QRDR) [15]. Eleven and eight of 27 clinical isolates, respectively, harbored *bla*_CTX-M-1_ and *bla*_CTX-M-9_, which, as extended-spectrum beta-lactamases, confer resistance to third-generation cephalosporins. Ceftriaxone MICs were positively correlated with the number of acquired resistance genes to beta-lactams in these isolates (i.e., the more *bla* resistance genes, the higher the MIC; ρ=0.66, *p* < 0.001). Moreover, three isolates (ST69 and ST410) carried *bla*_NDM-1_, which is responsible for carbapenem resistance.

Aiming to explore the impacts of AMR and AMR genes on morphological changes, the 28 *E. coli* isolates (27 clinical isolates and the reference strain ATCC 25922) were subjected to antimicrobial exposure for three hours at CLSI breakpoint concentrations [16] and examined by light microscopy. The microscopy images were qualitatively classified into four principal categories: lysis, ellipsoid, rod-like shape, and elongation (Figure 1a). Figure 1b demonstrates that *E. coli* cells responded to antimicrobial exposure with differential modifications in their morphology, with some showing an association with a particular AMR phenotype. Specifically, ceftriaxone-susceptible and -intermediate resistant isolates were elongated, while resistant isolates retained a rod-like shape when exposed to ceftriaxone (Figure 1b). Similarly, the majority of *E. coli* that were resistant to azithromycin, tetracycline, and gentamicin retained a rod morphology, while the majority of susceptible organisms were ellipsoid when exposed to these antimicrobials (Figure 1b).

Upon exposed to azithromycin and tetracycline, morphological changes were more heterogeneous and not consistent with the respective susceptibility profiles of the organisms. For example, of eight tetracycline-resistant *E. coli* isolates, five were ellipsoid and three were rod-like shape. Ciprofloxacin exposure induced the highest level of morphological heterogeneity, in which resistant organisms were rod-like shape, elongated, or a rod–elongated mixture; ciprofloxacin-susceptible organisms were ellipsoid, rod-shaped, or lysed when exposed to ciprofloxacin. We did not observe any obvious association between AMR gene content and bacterial morphological changes under exposure to antimicrobials (Figure 1b), with the exception of the aforementioned rod morphology with *mphA*, *aac(3)-IIa* gene, and *tet* gene variants conferring resistance to azithromycin, gentamicin, and tetracycline, respectively.

### 2.2. Assessing the Growth Dynamics of E. coli under Exposure to Antimicrobials

The 28 *E. coli* isolates were exposed to different concentrations (0 to 256 mg/L) of ciprofloxacin, azithromycin, ceftriaxone, tetracycline, and gentamicin; growth curves were performed over 24 h. One isolate was chosen to perform three technical replicates (Appendix A) and the pair-wise correlation coefficients among three replicates were calculated. An average correlation coefficient of 0.89 was determined. Additionally, each isolate was exposed to azithromycin and tetracycline with an additional one–two biological replicates to confirm the observed effects of resistance on their growth dynamics.

More than 2500 growth curves from the 28 *E. coli* isolates in different antimicrobial exposure conditions were generated, and Gompertz’s, Baranyi’s, and the three-phase linear models were fit to the growth curves. Gompertz’s and Baranyi’s models fit with the lowest MSE (mean square error) and MAE (mean absolute error), and the highest ρ (correlation coefficient) and $R^{2}$ (correlation of determination) (Appendix A). However, the performance of Gompertz’s model was superior to that of Baranyi’s model, which indicates that Gompertz’s (with three parameters) was the most suitable model for these data. Consequently, we employed Gompertz’s model to estimate the growth parameters of each curve for further analysis.

The growth features were plotted separately for five experimental conditions, and against different antimicrobial concentrations as relative to the isolate’s respective MIC (Figure 2). Generally, antimicrobials suppressed the growth of *E. coli,* which could be observed as the inferred A reduced gradually and the area under the curve (AUC) reduced rapidly with increasing drug concentrations. Alternatively, increased antimicrobial concentrations induced a prolonged λ in all cases. The μ against relative drug concentration was more heterogenous and largely dependent on antimicrobial class. The μ of *E. coli* treated with ciprofloxacin and tetracycline declined inversely with respect to an increase in the respective antimicrobial concentrations. However, when organisms were exposed to ceftriaxone, the μ remained constant, with some organisms showing increased μ at higher concentrations. Notably, under azithromycin exposure, the general trend of the 28 *E. coli* isolates was that μ increased in lower concentrations of the macrolide and then decreased gradually at higher drug concentrations. These data suggest that sub-inhibitory concentrations of azithromycin induce an increased rate of *E. coli* replication.

On further investigation, we also found that tetracycline-resistant isolates possess a fitness advantage over susceptible isolates in both antimicrobial-free conditions and at 0.5 × MIC (Figure 3). Specifically, at 0.5 × MICs of tetracycline, the μ (*p* < 0.0001), A (*p* < 0.001), and AUC (*p* < 0.0001) of the tetracycline-resistant *E. coli* were all significantly higher than those of the susceptible organism (Figure 3). Without antimicrobial exposure, the AUC of tetracycline-resistant isolates was significantly higher than that of susceptible isolates (*p* < 0.05). Conversely, azithromycin-susceptible *E. coli* in drug-free conditions had a fitness advantage over azithromycin-resistant *E. coli* with significantly higher AUCs (*p* < 0.05) (Figure 3).

### 2.3. Effects of Azithromycin and Tetracycline Resistance on Growth Dynamics of E. coli

To further investigate the impact of azithromycin and tetracycline resistance on the growth dynamics of *E. coli*, we plotted the growth features of resistant and susceptible isolates at different relative concentrations of these antimicrobials. Figure 4 shows a comparison between trends of growth features (as inferred by Gompertz’s model) on data from three replicates of resistant and susceptible organisms grown in increasing concentrations of azithromycin and tetracycline. We found that the λ of the tetracycline-resistant organism was lower than that of the susceptible organisms, while the μ and AUC of organisms resistant to tetracycline were higher than those of susceptible organisms. Conversely, the azithromycin-resistant organisms exhibited a longer λ, a higher A, and a lower μ than the azithromycin-susceptible organisms. Additionally, we found that the μ of both azithromycin-resistant and -susceptible *E. coli* peaked at low antimicrobial concentrations relative to MIC, which was supported by both Gompertz’s, Baranyi’s, and the three-phase linear models (Appendix A). These data suggest that sub-inhibitory concentrations of azithromycin actually accelerate the growth of both susceptible and resistant *E. coli*.

### 2.4. Elevation of Growth Rate in ermB^-^ E. coli at Sub-MIC Concentrations of Azithromycin

All azithromycin-resistant *E. coli* isolates (*n* = 19) carried *mphA*; additionally, seven harbored *ermB* and one isolate harbored *lnuF*. In order to investigate the influence of *ermB*, which was prevalent among *E. coli* resistant to azithromycin, on the growth features of *E. coli*, we compared the growth features of the *E. coli* with the presence (*ermB^+^*) and absence (*ermB^-^*) of this gene. The μ, A, and AUC decreased in both *ermB^+^* and *ermB^-^* isolates, while λ was elevated with increased relative concentrations of azithromycin (Figure 5a). We found that *ermB^+^ E. coli* exhibited a higher A and AUC, shorter λ, and lower μ than those of *ermB^-^ E. coli.* At low concentrations of azithromycin relative to MICs, μ in *ermB^-^* isolates increased rapidly (peaked at ~0.03× MIC) and then gradually decreased. The highest μ value from the *ermB^+^* organisms was lower than that from *ermB^-^* organisms, which peaked at a greater relative concentration of azithromycin. We further found that at lower concentrations of azithromycin (absolute concentrations of 1 and 2 mg/L), the *ermB^-^* organisms exhibited significantly higher μ than that of *ermB^+^* organisms (Figure 5b), which was in concordance with data generated with relative azithromycin concentrations. Lastly, we found that MICs of *ermB^+^* isolates were significantly higher than those of *ermB*^-^ isolates (*p* < 0.05) (Figure 5c). Our data show that MICs against azithromycin are associated with the accumulation of the macrolide resistance gene.

## 3. Discussion

This study systematically assessed the physiological and morphological responses of AMR *E. coli* to clinically relevant antimicrobials. Regarding morphology, we found a relationship between resistance to ceftriaxone and cell elongation, which has been previously reported [17,18,19,20,21,22] and also utilized to rapidly detect resistant isolates [10,11]. However, we found that the molecular basis of AMR was not associated with antimicrobial-induced morphological responses. This observation suggests that morphology-based rapid antimicrobial susceptibility testing may be performed without prior knowledge of resistance mechanisms, which is a drawback of nucleic acid-based and protein-based methods. A limitation of morphology-based testing is the highly diverse dependence of morphological changes on the types of antimicrobial, even for antimicrobials belonging to the same family. Considering the β-lactams as an example, meropenem induces a spindle morphology in *E. coli*, while moxalactam induces filamentous cells [23]. Thus, for each antimicrobial, a prior imaging dataset of both resistant and susceptible organisms is required to infer the susceptibility of newly isolated organisms. For other antimicrobials used in this study, the morphology–resistance correlation was less consistent, which may be addressed by additional time data provided by live-cell imaging, which may be required to build highly accurate antimicrobial susceptibility tests [11,24]. The heterogenous responses to CIP, AZI, TE, and CN might be explained by the highly diverse genetic background of the *E. coli* collection comprising of 11 different STs. Our results showed antimicrobial-induced morphological changes in both resistant and susceptible isolates, suggesting that bacterial cells respond to drug pressure despite the resistance to antimicrobials. This is probably due to the residue, in the cytoplasm, of active antimicrobials which is not yet extruded by efflux pumps (e.g., TetA and TetB) or inactivated by enzymes (e.g., β-lactamases). This could also be explained by the weak interaction between antimicrobials (e.g., CIP) and modified target proteins (e.g., mutated GyrA).

Due to improved manufacturing technology, the price of tetracyclines has decreased dramatically, to the extent where they are the cheapest antimicrobials available. This has made tetracycline the second most widely used antimicrobial worldwide [25]. Although the usage of tetracyclines in human health is becoming less common, they are still important antimicrobials disease prevention and growth promotion in animals [26]. Our data demonstrated a high prevalence of tetracycline-resistant *E. coli* via the *tet* gene, which is in accordance with previous studies [27]. By comparing growth features of *E. coli* treated with tetracycline, we found that tetracycline-resistant isolates possessed a fitness advantage over susceptible isolates. This fitness advantage in the absence of the antimicrobial may explain the ubiquity of *tet* gene carriage in the gut microbiota of children and the sustained persistence of tetracycline resistance [27]. Our observations concur with a pharmacodynamic model [28], in which the fitness of resistant *E. coli* did not differ from susceptible *E. coli* in antimicrobial-free conditions. However, a recent study using growth curve analysis found that tetracycline resistance may exert a fitness cost of up to 10% on *E. coli* compared with a wild-type strain [29].

Azithromycin has been listed by WHO as one of the critically important antimicrobial reagents for human medicine [30]. Moreover, WHO have been developing guidelines to prophylactically prescribe azithromycin to infants and children to reduce child mortality [31], which might increase the resistance against macrolide antimicrobials globally [32]. Therefore, the molecular mechanisms and prevalence of azithromycin resistance have been studied extensively [3,31,33,34,35]. However, the influence of azithromycin exposure on the physiology of bacteria is not well understood. Here, we observed an accelerated μ of both resistant and susceptible isolates under exposure to low concentrations of azithromycin, raising a concern about potential impacts of this antimicrobial on developing new resistant mutants. These results suggest further investigations are required to evaluate the potential benefits of presumptive antimicrobial treatments compared to the risk of AMR development. We speculate that at sub-MIC concentration, azithromycin molecules inhibit protein synthesis by binding to the P site on the 50S subunit of the bacterial ribosome [36], inducing up-regulation of ribosome synthesis which is associated with higher μ [37,38]. With increased concentration of the macrolide, more ribosomal molecules are bound by azithromycin, which leads to insufficient protein synthesis and reduction in μ.

Notably, the trend of μ in azithromycin-resistant *E. coli* was lower than the trend in azithromycin-susceptible isolates. This observation may be experimental evidence supporting a mathematical model for ribosome-targeting antimicrobials, which showed that faster growth induces greater drug susceptibility [39]. Moreover, amongst the azithromycin-resistant *E. coli* harboring the *mphA* gene, the *ermB^+^* isolates exhibited lower μ than that of *ermB^-^*. We hypothesize two feasible causal pathways of how *ermB* influences μ: (1) the presence/absence of *ermB* directly impacts on the μ trend or (2) the presence of *ermB* increases the MICs of *E. coli* and consequently changes the trend of μ against the relative concentration of azithromycin. The lower μ in *ermB^+^* may be due to reduced ribosome synthesis as *ermB* allows constitutive methylations of 23S rRNA [40], which protects the complexes from azithromycin binding and potentially interferes with ribosome assembly. This suggests that the presence/absence of *ermB* is likely to have a direct impact on μ.

This study has limitations. First, using these approaches, we were unable to develop an algorithmic tool to quantitatively measure morphological changes and to categorize cell shapes. Therefore, the cell shape classification may be subjective. Second, the relationship between morphology, growth dynamics, and cell viability was not investigated. Third, the correlation between the live cell counts, measured CFUs, and OD values, representing the density of both live and dead cells, was not verified. Although bacterial enumeration provides information on the actual growth rates of the cultured bacteria, we opted to analyze the growth of *E. coli* by monitoring OD as it enables high-throughput experiments with biological replications. Lastly, the 27 *E. coli* isolates possessed potential cross-interaction among AMR genes and high diversity in genetic background, which we were not able to address.

## 4. Materials and Methods

### 4.1. Organisms and Whole Genome Sequencing

Twenty-seven *E. coli* isolates were isolated from the blood samples of Vietnamese children hospitalized with sepsis in a previous study [41], which was approved by the Oxford Tropical Research Ethics Committee (OxTREC, reference number 35–16) and the Ethic Committee of Children’s Hospital 1 (reference number 73/GCN/BVND1). *E. coli* ATCC 25922 was included as a control in all analyses. Minimum inhibitory concentrations (MICs) against ciprofloxacin, azithromycin, ceftriaxone, tetracycline, and gentamicin were determined for all isolates by E-test (bioMerieux, Ivry-sur-Seine, France) and serial broth dilution. Whole-genome sequencing was performed on the 27 clinical isolates. DNA was extracted and purified using the Wizard Kit (Promega, Madison, WI, USA); the Nextera XT DNA Library Preparation Kit (Illumina, San Diego, CA, USA) was used to fragment DNA with adapter ligation. DNA sequencing was performed on a MiSeq platform, using MiSeq Reagent Kit V2 (Illumina, San Diego, CA, USA) to produce 250-bp paired-end reads. Sequencing data are available at www.ebi.ac.uk/ena (accession numbers from ERR4342165 to ERR4342191).

### 4.2. Monitoring and Modeling Bacterial Growth

For growth dynamics experiments, two-fold serial broth dilutions were conducted in 96-well U-bottom plates (Thermo Fisher Scientific, Waltham, MA, USA) to obtain differing concentrations of antimicrobials (from 0 to 512 mg/L) in 100 μL cation-adjusted Mueller-Hilton (MH) broth (Oxoid, Basingstoke, UK). The selected antimicrobials were ciprofloxacin, azithromycin, ceftriaxone, tetracycline, and gentamicin (Sigma Aldrich and AK Scientific, MO, USA). For each of the 28 *E. coli* isolates, one colony of overnight-incubated *E. coli* was inoculated into 5 mL Luria-Bertani (LB) broth (Basingstoke, UK) for 3 h at 37 °C with shaking to reach OD_600_ ~ 0.3. The bacterial suspension was diluted 1/1000 in MH broth, and 100 μL of the diluted suspension was mixed into each well of the 96-well plates. The final concentration of antimicrobials in each well varied from 0 to 256 mg/mL (two-fold intervals, i.e., 0, 0.5, 1, 2, …, 256). OD_600_ was measured on the plates every 15 min for 24 h at 37 °C, with shaking before each measurement, in a FLUOstar OPTIMA microplate reader (BMG Labtech, Ortenberg, Germany). The output data for each condition were curated and modified Gompertz’s (1), Baranyi’s (2), and three-phase linear (3) models [13] were fit to the data by minimizing the mean square error (MSE).
(1)$$N\left(t\right)=N_{0}+\mathrm{Aexp}\left\{-\exp\left[\frac{\mu e}{A}\left(\lambda -t\right)+1\right]\right\}$$
(2)$$N\left(t\right)=N_{0}+\mu C\left(t\right)-\ln\left\{1+\frac{e^{\mu C\left(t\right)}-1}{e^{A}}\right\}$$
(3)$$N\left(t\right)=\{\begin{array}{ccc} N_{0} & ,\;\; & t\leq \lambda \\ N_{0}+\mu \left(t-\lambda \right) & ,\;\; & \lambda <t<t_{s}\\ A & ,\;\; & t\geq t_{s} \end{array}$$
where N(t) is the cell density, measured in OD_600_, at timepoint t, $N_{0}$ is the cell density at t=0, A is the maximum cell density, μ is the maximum growth rate (per hour), λ is the lag time (in hours), $t_{s}$ is the time to reach stationary phase (in hours), and $C\left(t\right)=t+\frac{1}{\mu }\ln\left[e^{-\mu t}+e^{-\mu \lambda }-e^{-\mu \left(t+\lambda \right)}\right]$. Area under the curve (AUC) was approximately estimated from the growth curves using the Reimann summation method. We termed the A, μ, λ, and AUC as growth features of a growth curve.

### 4.3. Bright-Field Microscopy

In order to investigate the morphology of *E. coli* in response to antimicrobial exposure, isolates were inoculated on a nutrient agar (NA) plate and incubated overnight at 37 °C. For each isolate, <10 colonies were inoculated into saline buffer (0.85% NaCl) to reach OD_600_ ~ 0.6. Resistant isolates were diluted 1/20 to reduce cell density in a field of view under microscopic examination. The selected antimicrobial (ciprofloxacin, azithromycin, ceftriaxone, tetracycline, and gentamicin) of its corresponding CLSI break-point concentration [16] (i.e., ceftriaxone at 4 μg/mL) was added to the bacterial suspension and to the thin NA plate (8 mL). Then, 100 μL of bacterial suspension was inoculated onto NA plates and incubated for 3 h at 37 °C. Subsequently, small pieces of agarose pad were excised from the NA plate, placed onto microscope slides, and examined under a Nikon Ni-e upright microscope with a 100× objective lens.

### 4.4. Data Analysis

Data analysis was performed in R (version 3.5.3) programming language [42]. To estimate the parameters from the models, the “optim” function (the R Stats package) was used to constrainedly minimize the MSE between the actual and predicted OD values. For each isolate, the growth features were plotted against relative MICs using the “ggplot2” and “gridExtra” packages. Wilcoxon rank-sum test on boxplots was carried out on the “ggpubr” package. Multilocus sequence typing (MLST) of *E. coli* was inferred from the whole-genome sequencing data using the SRST2 pipeline [43] on the seven housekeeping genes, including *adk*, *fumC*, *gyrB*, *icd*, *mdh*, *purA*, and *recA*. Resistome analysis with AMR gene identification was performed on the SRST2 pipeline. De novo assembly of sequencing data was performed using SPAdes [44], gene annotation was completed using Prokka [45], and the pan- and core genomes were deduced using Roary [46] with a minimum 99% blastp identity. Finally, a phylogenetic tree was constructed on RAxML [47] using the alignment of the core genome from Roary’s output and was visualized using the “ggtree” package.

## 5. Conclusions

In conclusion, we exploited growth dynamics and bacterial morphology to study the impact of AMR in clinical *E. coli* isolates. The morphological responses to ceftriaxone were consistently associated with the resistance to the antimicrobial, while the responses to other antimicrobials were more heterogeneous. In the absence of the drug, tetracycline resistance confers a fitness advantage on *E. coli*, while the other antimicrobials exert a measurable fitness cost (lower AUC and μ) on these bacteria. At low concentrations of azithromycin, μ was accelerated and azithromycin-susceptible isolates possessed higher μ than resistant isolates. We additionally observed an impact of AMR genes on the growth dynamics of bacteria, in which *E. coli* harboring the *ermB* gene exhibited lower μ than *ermB^-^ E. coli*. Our study raises a concern regarding the persistence of AMR and the development of new resistance phenotypes under antimicrobial exposure, showing the need for a better understanding of how morphological and physiological responses contribute to the AMR and tolerance in bacterial pathogens.

## Figures and Tables

**Figure 1 antibiotics-09-00735-f001:**
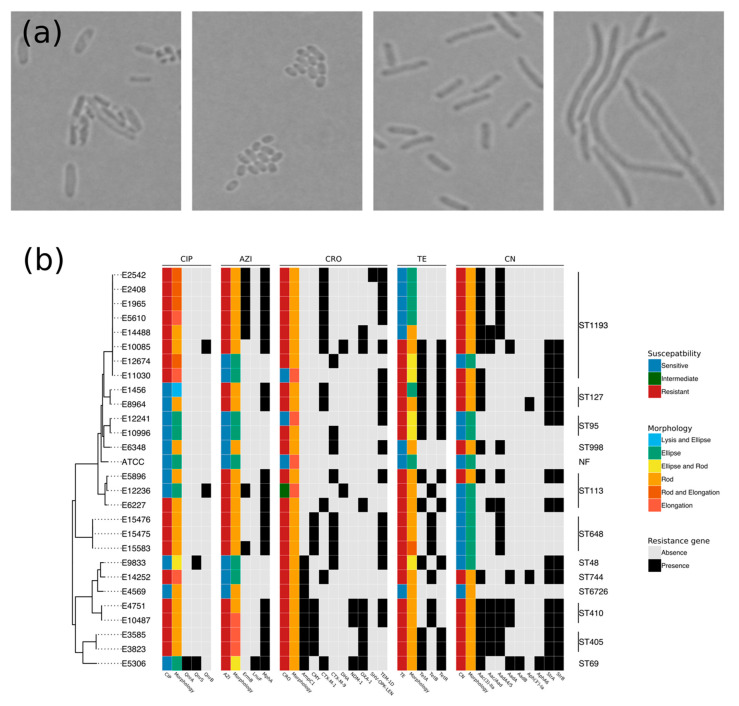
Bacterial morphology associated with antimicrobial resistance (AMR) and resistome. (**a**) The morphological features of *E. coli* after 3 h of being treated with ciprofloxacin. Brightfield images of *E. coli* in different morphologies, including, from left to right, cell lysis, ellipsoid, rod-like shape, and elongation, were acquired with a Nikon Ni-E upright microscope and a 100× objective lens. Morphology of the reference strain is in rod-like shape in drug-free condition. (**b**) The phylogenetic structure of *E. coli* with corresponding susceptibility, AMR genes, and morphology. (Left) The unrooted phylogenetic tree constructed from core genomes of the 28 *E. coli* isolates. (Right) Sequence type (ST) of those isolates obtained by mapping the whole-genome reads to a database of seven housekeeping genes. (Middle) Five heatmap blocks, each of which demonstrates (first column) antimicrobial susceptibility, (second column) morphology under exposure to antimicrobials, and (remaining columns) the presence (black) and absence (gray) of resistance genes against quinolone (CIP: ciprofloxacin), macrolide (AZI: azithromycin), β-lactam (CRO: ceftriaxone), tetracycline (TE), and aminoglycoside (CN: gentamicin). NF: non-identified ST.

**Figure 2 antibiotics-09-00735-f002:**
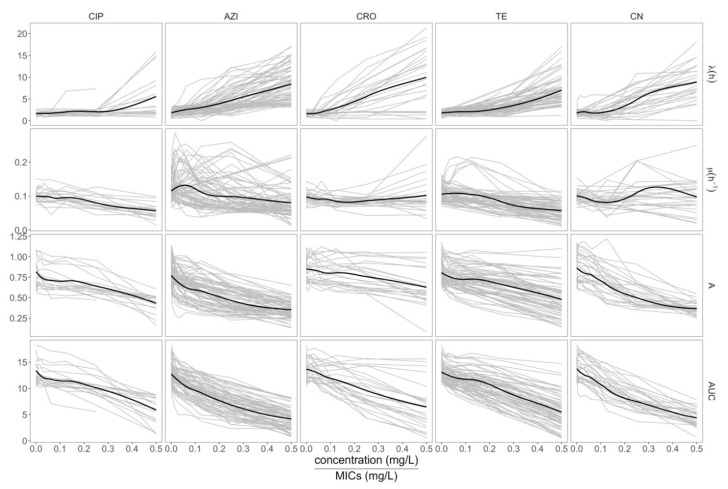
The dynamics of the growth features. From top to bottom, lag-phase period (λ), maximum growth rate (μ), maximum cell density (A), and area under the curve (AUC) corresponding to antimicrobial concentration relative to MICs of 28 *E. coli* isolates from clinical samples. From left to right, ciprofloxacin (CIP), azithromycin (AZI), ceftriaxone (CRO), tetracycline (TE), and gentamicin (CN). The horizontal axis is antimicrobial concentrations relative to MICs of each isolate. The vertical axis represents the values of growth features. Bold lines are loess smooth functions and thin lines are the growth features of each isolate.

**Figure 3 antibiotics-09-00735-f003:**
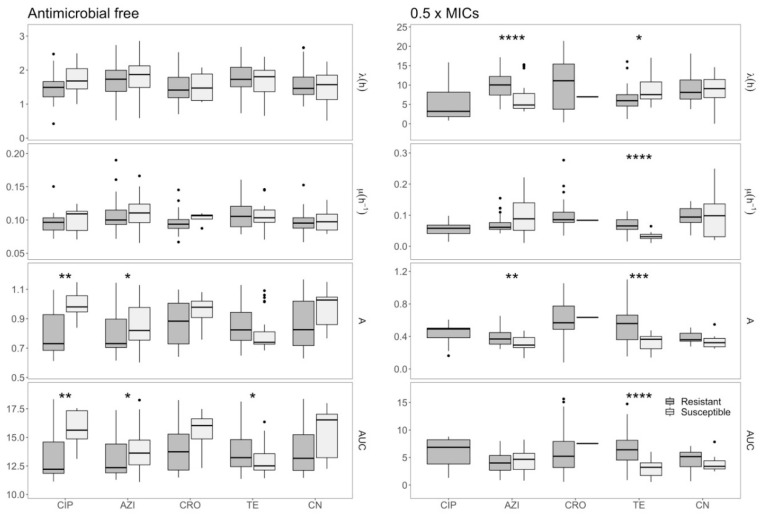
A comparison between the growth features for susceptible isolates and those for resistant isolates. From top to bottom, maximum cell density (A), maximum growth rate (μ), lag-phase period (λ), and area under the curve (AUC) of resistant isolates (dark gray) and those of susceptible isolates (light gray) without antimicrobial (left) and at half MICs (right). *: *p*-value < 0.05, **: *p*-value < 0.01, ***: *p*-value < 0.001, ****: *p*-value < 0.0001.

**Figure 4 antibiotics-09-00735-f004:**
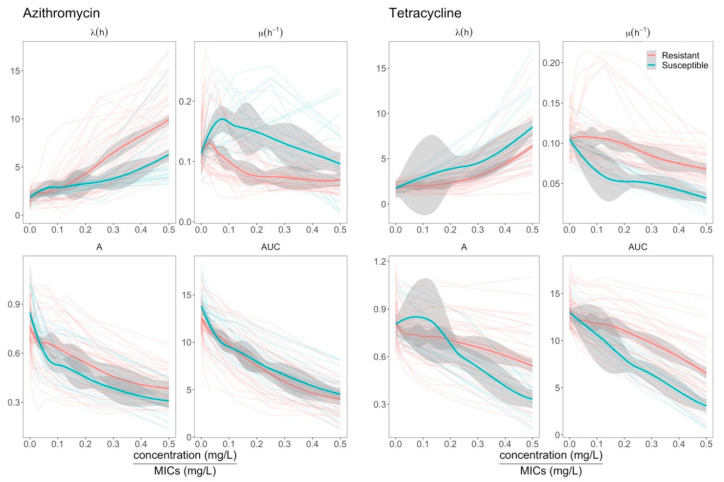
A comparison between growth features of resistant and susceptible isolates under exposure to azithromycin and tetracycline. The horizontal axis is antimicrobial concentrations relative to MICs. The vertical axis is values of the four growth features, including lag time (λ), maximum growth rate (μ), maximum cell density (A), and area under the curve (AUC). For each isolate, growth features were plotted against relative concentrations of antimicrobials as thin lines. Bold red (resistant isolates) and blue (susceptible isolates) lines indicate loess smooth regression. Gray shading represents 95% confident intervals.

**Figure 5 antibiotics-09-00735-f005:**
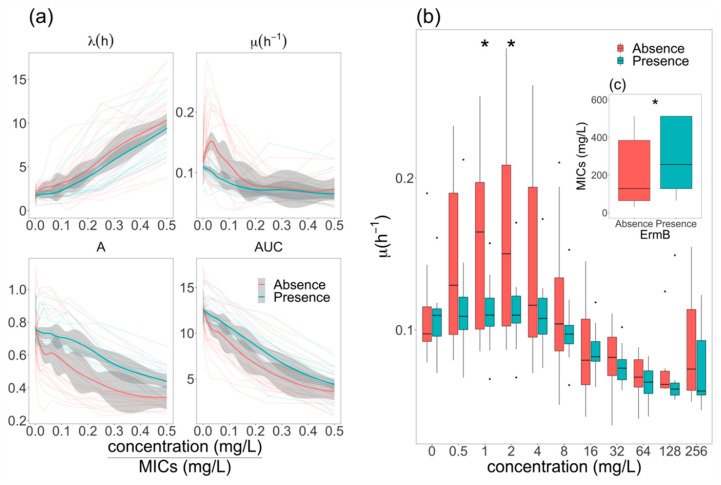
The impact of the *ermB* gene on growth dynamics of azithromycin-resistant *E. coli***.** (**a**) Growth features of *E. coli* in the presence (blue thin lines) and absence (red thin lines) of *ermB*. Vertical axis is values of λ: lag time, μ: maximum growth rate, A: maximum cell density, and AUC: area under the curve. Horizontal axis is antimicrobial concentration relative to MICs. Bold red and blue lines are loess smooth regression. Gray shading is 95% confident intervals. (**b**) Maximum growth rate (μ) of *ermB^-^ E. coli* (red boxes) compared to that of *ermB^+^ E.coli* (blue boxes). Horizontal axis is the absolute concentration of azithromycin (mg/L). (**c**) Comparison MICs of *ermB^+^ and ermB^-^ E. coli*. *: *p*-value < 0.05.

**Table 1 antibiotics-09-00735-t001:** Minimum inhibition concentrations (MICs) and antimicrobial resistance of ATCC 25922 and 27 clinical isolates used in this study.

Isolate ID	Minimum Inhibition Concentrations (mg/L)				
	CIP ^1^	AZI ^1^	CRO ^1^	TE ^1^	CN ^1^
E3585	32 (R)	48 (R)	256 (R)	256 (R)	256 (R)
E3823	32 (R)	48 (R)	256 (R)	256 (R)	256 (R)
E4569	0.064 (S)	12 (S)	32 (R)	4 (S)	1 (S)
E6348	0.38 (S)	8 (S)	256 (R)	4 (S)	256 (R)
E8964	0.38 (S)	64 (R)	256 (R)	256 (R)	64 (R)
E9833	0.38 (S)	16 (S)	256 (R)	256 (R)	1 (S)
E10085	32 (R)	48 (R)	256 (R)	16 (R)	256 (R)
E10487	32 (R)	256 (R)	256 (R)	256 (R)	96 (R)
E10996	0.094 (S)	16 (S)	256 (R)	256 (R)	1 (S)
E11030	32 (R)	4 (S)	0.094 (S)	256 (R)	64 (R)
E12236	0.25 (R)	256 (R)	1.5 (I)	256 (R)	1 (S)
E12241	0.094 (S)	8 (S)	0.064 (S)	128 (R)	1 (S)
E12674	32 (R)	6 (S)	64 (R)	128 (R)	1.5 (S)
E14252	12 (R)	16 (S)	256 (R)	256 (R)	32 (R)
E14488	32 (R)	256 (R)	256 (R)	4 (S)	16 (R)
E15475	32 (R)	256 (R)	256 (R)	256 (R)	1 (S)
E15476	32 (R)	256 (R)	256 (R)	256 (R)	1.5 (S)
E15583	32 (R)	256 (R)	256 (R)	256 (R)	1 (S)
E1456	0.25 (S)	48 (R)	256 (R)	64 (R)	256 (R)
E1965	32 (R)	256 (R)	256 (R)	4 (S)	256 (R)
E2408	32 (R)	256 (R)	256 (R)	4 (S)	64 (R)
E2542	32 (R)	256 (R)	256 (R)	4 (S)	256 (R)
E4751	32 (R)	256 (R)	256 (R)	256 (R)	256 (R)
E5306	0.38 (S)	48 (R)	256 (R)	256 (R)	256 (R)
E5610	32 (R)	256 (R)	256 (R)	4 (S)	256 (R)
E5896	0.38 (S)	256 (R)	128 (R)	256 (R)	256 (R)
E6227	32 (R)	256 (R)	256 (R)	256 (R)	1.5 (S)
ATCC25922	0.094 (S)	4 (S)	0.125 (S)	4 (S)	1.5 (S)
Susceptible	8/27 (29.6%)	8/27 (29.6%)	2/27(7.4%)	7/27 (25.9%)	10/27 (37.0%)
Intermediate	0/27 (0.0%)	0/27 (0.0%)	1/27 (3.7%)	0/27 (0.0%)	0/27 (0.0%)
Resistant	19/27 (70.4%)	19/27 (70.4%)	24/27 (88.9%)	20/27 (74.1%)	17/27 (63.0%)

^1^ CIP: ciprofloxacin, AZI: azithromycin, CRO: ceftriaxone, TE: tetracycline, CN: gentamicin, S: susceptible, I: intermediate, R: resistant.

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
