# Peer review of "Pathogenic Escherichia coli Possess Elevated Growth Rates under Exposure to Sub-Inhibitory Concentrations of Azithromycin"

_antibiotics, 2020, doi:10.3390/antibiotics9110735_

Round 1

Reviewer 1 Report

The authors exploited growth dynamics and bacterial morphology to study the impact of AMR in clinical E. coli isolates. The morphological responses to ceftriaxone associated with resistance. Responses to other antimicrobials was more heterogeneous. In the absence of the drug, tetracycline resistance confered a fitness advantage while the other antimicrobials exerted a fitness cost. AMR genes like ermB impact growth dynamics of bacteria. The persistence of AMR under sub-MIC antimicrobial exposure is poorly understood with important clinical implications.

The paper is very well written and raises an important set of issues. My only main issue is that the approach the paper takes, despite the mathematical treatment of the data, is phenomenological. I think the authorss should make a more reasonable effort to provide molecular explanations in their discussion as to why sub-MIC concentrations of antibiotics have such effects and why they think that different antibiotics have different effect.s

Major editing

  1. First, using these approaches we were unable to develop an algorithmic tool to quantitatively measure morphological changes and to categorize cell shapes. Therefore, the cell-shape classification may be subjective.

this is an example of a general issue I have with the paper. There is little connection to the molecular basis of. the phenomena studied. I think there needs to be more molecular discussion that can bee discussed as possible explanation to the phenomena studied.

Author Response

As the reviewer suggested, extra discussion on molecular explanation of observed phenomena in this study has been included accordingly in the main text at line 254-261 and line 288-292.

Reviewer 2 Report

This study explores the morphological and physiological responses of a number (27) of clinical isolates of E. coli under short-term exposure to 5 antimicrobials (ciprofloxacin, ceftriaxone, tetracycline, gentamicin, and azithromycin) for finding out possible correlations among morphology and antimicrobial resistance (AMR). All E. coli isolates were submitted to whole genome sequencing to evaluate their AMR gene content.

It is found that AMR is common, with all organisms resistant to at least one antimicrobial and most of than were multi-drug resistant (MDR). Results outline complex interactions between E. coli and antimicrobials. Usually, exposure to antimicrobial suppressed bacterial growth, but sub-MIC concentrations of azithromycin increased the growth rate of the clinical isolates. This raises clinical concerns regarding exposure of sub-MIC concentrations of specific antimicrobials.

Regarding morphology, a relationship between resistance to ceftriaxone and cell elongation is found, but for other antimicrobials used in this study, the morphology-resistance correlation was less consistent. In general, the molecular basis of AMR was not associated with antimicrobial-induced morphological responses.

Anyway, the paper is interesting. According the WHO AMR is one of the ten major threats to global heath and this point is enough to justify the study. Introduction is nicely focused and methods are appropriate. Mathematical models for studying bacterial growth are also convenient. Discussion about limitations of the study (Lines 285-294) is very interesting and it is a contribution and a guide for future research.

The following `points should be addressed before definitive publication

According to heading in Table 1, the number of 28 isolates is justified: ATCC 25922 and 27 clinical isolates used in this study. However, throughout the text, the number of isolates is referred sometimes 27 and in other paragraphs 28 (see lines 102, 109 but also lines 165). This is confusing and it should be unified. The AYCC should be always included or not

Line 118: it is stated that the majority of susceptible organisms were ellipsoid when exposed to these antimicrobials (Figure 1). Later, at lines 500, wild type are rod-like... so, is wild type susceptible to antimicrobials? Please, clarify

Distribution of Figures in the regular and supplementary material is not clear. I think that according the goal of the study, Figures S1 is simple and illustrative, so that this figure would be incorporated to the paper. On the other hand, Figure 4 seems to be m0ore appropriated for the supplementary material.

Author Response

The reference strain ATCC 25922 was included to all analysis of this study (line 334). Clarification has been made accordingly in line 88-90 and line 117 The number of isolates in line 172 was corrected as 28.

The wild type was susceptible to all five antimicrobials used in this study. The wild type strain has been corrected as reference strain in line 107. Accordingly, we have clarified the rod-like morphology of the strain in drug-free condition.

The figure S1 has been combined into figure 1 in the main text accordingly. Figure 4 has remained in the main text since it supports a major result in section 2.3.

Reviewer 3 Report

Dear authors, as you underlined in lines from 285 to 294 the study has important limitations and the results, you show, only underline the importance of not exposing E. coli to sub-MIC concentrations of specific antimicrobials. 

You should improve these limitations and then you could resubmit the paper. 

Author Response

We do apricate your insightful comments. The objectives of this work is to observe the morphological and physiological responses of clinical E. coli to different classes of antimicrobial which limited in-depth investigation in molecular mechanism of the observed phenomena. However, we have included extra hypothesis and speculation in discussion (line 254-261 and line 288-292) to explain the study results.

Reviewer 4 Report

In the manuscript, the authors introduced a broad investigation for the understanding of antibiotic-resistant in bloodborne E. coli from both a cellular and molecular perspective using microscopic techniques and whole-genome sequencing. Also, they analyzed the growth dynamics of pathogenic E. coli to measure the physiological changes with a diverse antibiotic-resistant gene upon short-term exposure to antibiotics. I believe that this research has shown the usage of dynamic analysis of the growth curve to understand the effect of antibiotic-resistant and its gene to bacterial growth and morphological change. However, the following issues are needed to be addressed before it can be published.

Major issues:

  1. In figure 1 and S1, the authors presented the morphology of the bacteria under antibiotic exposure. however, it is not clear at which concentration the morphology was determined. As the morphological change is dependent on the antibiotic concentration, the concentrations are needed to be presented. Was it over MIC or under MIC? In addition, as resistant bacteria to certain antibiotics, the morphology should not be changed from the original shape. However, there are morphology changes in even resistant strains. Please explain the procedure of the determination of morphology and explain the reason for a morphological change in resistant strains.

  1. The growth curve is generated by measuring the optical density of culture media in 96 well plates. As the author used the U-shape well plate, the bacterial cells would be concentrated at the bottom of the plate. The author shook the well plate before each measurement. What was the method of shaking? Manually or using a machine? As the density of cells in the culture plate increases with time, it would be not easy to make even distribution. How do you confirm that the bacterial cell distributed well? Was the well plate exposed to the room temperature during shaking and measurement? If so, it would affect the measurement. Please clarify these issues.

Minor issues

  1. in figure 2, the order of the figure and caption does not match. Please check the order of figures and captions.

Author Response

The bacterial morphology was obtained after 3h of exposure to antimicrobials at CLSI breakpoint concentrations (section 4.3), which could be either over or under the MICs of the strains in the studied collection. The procedure of morphology determination was described in section 4.3. We have accordingly explained the morphological changes in resistant isolates at line 254-261.

The 96-well U-bottom plates were incubated at 37oC and measured optical density every 15 minutes for 24 hours, with automatic shaking before each measurement (section 4.2). All steps of the procedure was performed automatically in the OPTIMA microplate reader.

The microplates were automatically shaken with maximum speed to ensure that the bacterial suspension in each well is as homogenous as possible. However, a minor amount of bacterial pellets remained at the bottom of each well after shaking, adding systematic error to OD measurement.

This systematic error did not alter the study results and conclusion since the error occurred in every measurement at all timepoints and drug concentrations.

As suggested by the reviewer, the figure legend was corrected.

Round 2

Reviewer 3 Report

No comments or suggestions

Reviewer 4 Report

I am happy with the responses and revised manuscript of the authors.